# Incorporating Traditional Knowledge into Science-Based Sociotechnical Measures in Upper Watershed Management: Theoretical Framework, Existing Practices and the Way Forward

Hunggul Yudono Setio Hadi Nugroho *[ID], Markus Kudeng Sallata, Merryana Kiding Allo, Nining Wahyuningrum, Agung Budi Supangat, Ogi Setiawan [ID], Gerson Ndawa Njurumana, Wahyudi Isnan, Diah Auliyani [ID], Fajri Ansari [ID], Luthfi Hanindityasari and Nardy Noerman Najib

Research Center for Ecology and Ethnobiology, National Research and Innovation Agency,
Jalan Raya Jakarta-Bogor Km.46, Cibinong 16911, Indonesia
* Correspondence: hunggul.yudono.setio.hadinugroho@brin.go.id

**Abstract:** In Indonesia, 2145 watersheds currently need to be restored, where around 21 million people spread over ± 23,000 villages live below the poverty line with a high dependence on forests. This condition requires an integrated approach in watershed management, which is aimed at technically restoring environmental conditions and ensuring the welfare of the people in it. One of the strategic approaches that can be taken is to revive local wisdom and traditional knowledge (TK), which has been eroded and neglected, and integrate them with technical approaches based on modern science and knowledge. Based on the author's research and literature studies, this paper discusses the theoretical framework and implementation practices in integrating traditional knowledge into a science-based sociotechnical system to manage upstream watersheds sustainably. Based on the empirical evidence, efforts to create good biophysical and socio-economic watershed conditions can only be achieved through the active participation of farmers in adopting and integrating scientific technology into their traditional knowledge. This integration is realized in designing and implementing watershed management technology by considering the principles of suitability, applicability, feasibility, and acceptability. In the long term, it is necessary to document TK, patent it, and transfer it to the next generation to ensure that indigenous peoples' and local communities' social, cultural, and economic interests are protected.

**Keywords:** norms; values and beliefs; evidence-based measures; sustainability

## 1. Introduction

The watershed has been accepted as a unit management with natural-based boundaries, where all the resources are integrated, and it is not only focused on land, water, and vegetation improvement but also on alleviating poverty. In Indonesia, about 37.2 million people distributed in approximately 25,000 villages live with close interaction with forests and are generally located in the upper watershed. This population consists of 9.2 million households, 1.7 million of which live under the poverty line [1].

The foremost vital factor that causes the poor performance of watershed management is the failure to link the perceptions, needs, abilities, and capacities of the local communities living in upper watershed areas with technical aspects and policy implementation approaches. Watershed management has often been merely based on a scientific knowledge-based technical approaches [2], which sometimes neglect local communities' capacities [3]. Therefore, it is essential to re-arrange these approaches in favor of sociotechnical system approaches that are strengthened by scientific knowledge. Watershed management requires a shift from a focus on technocratic aspects, where technical expertise is strengthened, to a range of social approaches, including improving institutional systems, and from a top-down, target-oriented approach to a more participatory, integrated,

and process-based approach. Traditional knowledge (TK) should be incorporated into science-based socio-technical systems [4]. Local people and their TK are central in every watershed management process [5]. Watershed degradation issues should be addressed based on local people's perspectives whilst at the same time being rationalized by scientific knowledge in the frame of the socio-technical system. This would aim to achieve a more constructive interaction between the social and technical systems, leading to boost in social innovation [6]. Every policy must consider the rights and needs of present and future generations of local people. Different community goals, namely social, economic, and environmental aspects, need to be either integrated or traded off if they are inappropriate.

This paper examined the TK and science-based socio-technical measures related to upper watershed management and incorporated these two dimensions to achieve sustainable management. This paper aimed to provide a theoretical framework and lessons learned in terms of best management practices in an upper watershed that integrates all dimensions whilst incorporating TK. The sources and materials of the review consisted of the author's publications and experiences, scientific publications, research reports, and other relevant materials.

## 2. Methods

This study was conducted based on a literature review and the author's research experience on TK and socio-technical systems in watershed management. This paper begins with an overview of the definition of TK, followed by an exploration of the TK that still exists in various parts of Indonesia, namely Nusa Tenggara, Java, Sulawesi, and Kalimantan. This sub-section also discusses the limitations and obstacles to applying TK. This overview was intended to equalize the perceptions and understanding of TK related to watershed resource management. The Section 4 discusses the theoretical framework of socio-technical systems in natural resource management. The Section 5 discusses a socio-technical approach to watershed management by integrating TK and science, starting from planning and implementation and proceeding to monitoring and evaluation activities. The Section 7 explains how to preserve the existing TK while adapting it to changing times and watershed management issues.

## 3. Traditional Knowledge in Watershed Management

### 3.1. Definition and Scope

Contextually, TK is knowledge that develops through an adaptation process from a community group into collective knowledge. It relates to a particular place and a long process time and is passed down from generation to generation. TK can also be interpreted as local cultures, such as how to achieve something in response to the environment based on collective past, present, and future experiences and knowledge [7,8]. TK, often called indigenous wisdom or knowledge, is rooted in cultural experience that guides the holistic relationship between human beings and their environment [9–11].

TK contributes to understanding natural processes and environmental properties and utilizes them for managing natural resources, adaptation, mitigation of hazards and disasters, and climate change [12–14]. TK has previously existed in the context of watershed management, particularly in the upper watershed. However, it is sometimes neglected in planning, implementing, and monitoring watershed management.

### 3.2. Recognition of Traditional Knowledge in a Watershed-Related Regulatory System

There have been several laws introduced in the last decade that mention community participation, including the Government Regulation No. 37 from 2012 concerning watershed management, the Law No. 37 from 2014 on soil and water conservation (SWC), and the three Ministry of Environment and Forestry (MoEF) Regulations No. 9 from 2021 on the management of social forestry, No. 23 from 2021 on the implementation of land and forest rehabilitation (LFR), and No. 10 from 2022 on the preparation and annual planning of forest and land rehabilitation.

The Law No. 37 from 2014, and the MoEF Regulations No. 9 from 2021, No. 23 from 2021, and No. 10 from 2022 mention the term local wisdom. In the Law No. 37 from 2014 concerning SWC, it is mentioned that land managed by customary law communities or traditional communities that carry out local wisdom is an exception from the obligation of every person who has land rights in protected and cultivated areas to carry out SWC activities. Furthermore, the MoEF Regulations No. 9 from 2021, No. 23 from 2021, and No. 10 from 2022 mention local wisdom as a basis for the consideration of planning and implementing social forestry and LFR.

### 3.3. Traditional Knowledge-Based Watershed Management Practices in Various Regions

Based on the population census of 2010, about 1340 ethnicities live in Indonesia [15]. They have unique cultures and traditions in terms of TK diversity in articulating natural resource management [16]. Below, we present the implementation of TK in natural resource management in four regions in Indonesia with different characteristics: Nusa Tenggara, Java, Sulawesi, and Kalimantan (Figure 1).

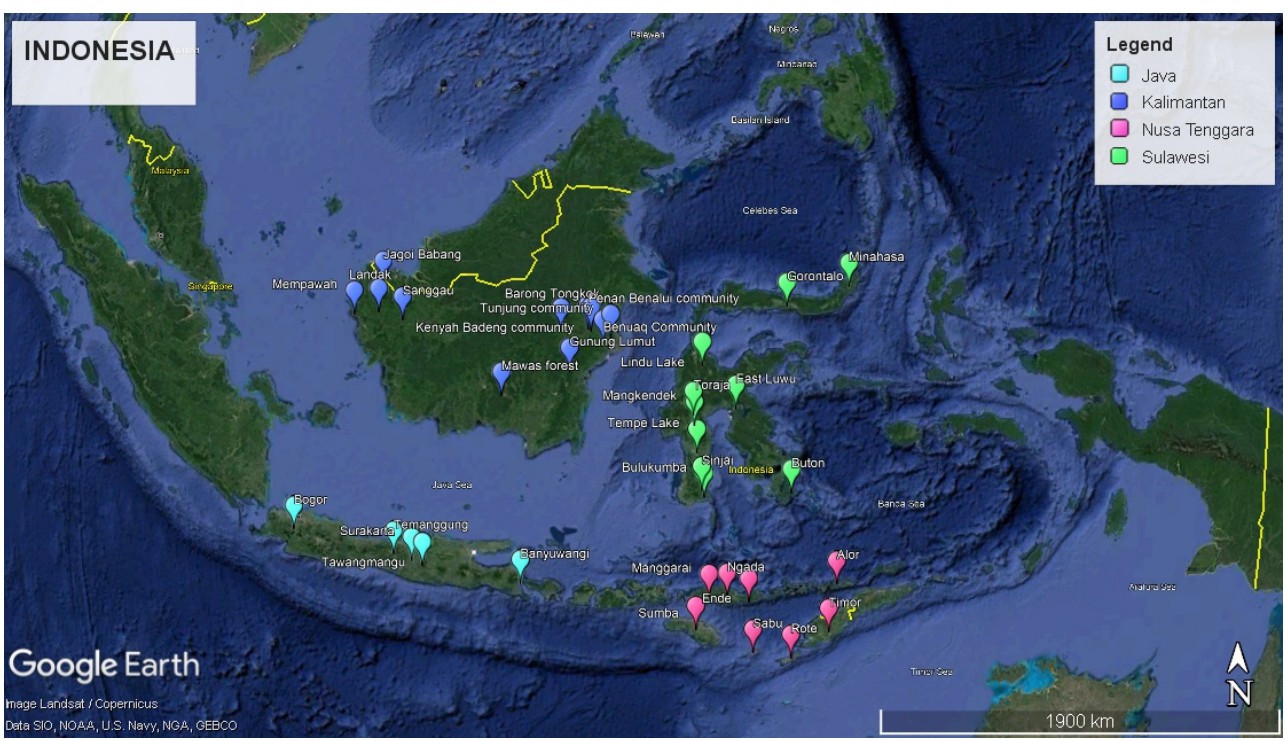

**Figure 1.** Distribution map of traditional knowledge locations in four major islands in Indonesia with different characteristics.

### 3.3.1. Nusa Tenggara: Small Islands with a Dry Climate and Low Population Density

West Nusa Tenggara (NTB) Province is dominated by the Sasak and Sumbawa ethnicities, while East Nusa Tenggara (NTT) Province is home to the Alor, Atnoni, Ende, Manggarai, Ngada, Sumba and Rote ethnicities. The TK of natural resource management of these provinces depends on biophysical conditions [17]. Generally, the watersheds in these regions are small, the water directly flows into the sea, there is a high watershed gradient, and the land is dominated by steep slopes, mountains, and hill landforms. The climate is relatively dry with a short rainy season period, and floods and erosion are common issues. The TK in this region is in the form of cultivation techniques, land system arrangement, and traditional regulations related to natural resources.

The indigenous knowledge in terms of managing land resources in the dryland region of NTT is quite varied. The Mutis community in the upper Benain–Noelmina watershed of Timor island has the concept of *mansian muitnasi nabua*, which means that humans, forests,

and livestock are an inseparable unit of life [18]. The local community utilizes landscape-based forest lands called *Suf* (traditional management areas based on clans/tribes for traditional ceremonies, harvesting honey, and raising livestock). The people on Timor Island are knowledgeable about sustainable farming in the local watershed ecosystem called *mamar*, which involves four main components, i.e., farmers, livestock, annual crops, and perennial trees [19]. The *mamar* farming system is practiced by 30–50% of the farmers in the region and contributes significantly to their food security and income as well as the socio-cultural prestige of the farmers [20].

A lot of the local wisdom of the Sumbanese community supports watershed conservation. *Mondu* is a farming system that implements soil and water conservation and vegetation cultivation to protect river borders from erosion and landslides [21]. Additionally, *kaliwu* (Figure 2) can be used as a model for hilly area conservation. *Kaliwu* also provides environmental services regarding community incomes, such as firewood, carpentry wood, food, and medicinal plants [22]. Another aspect of local wisdom is *lende ura*, which belongs to the community who live around Mount Yawila. Based on this local wisdom, they believe in the role of forests as rain bridges to support agriculture and the availability of water for humans and living things [23]. This knowledge promotes community participation in land rehabilitation and the preservation of the protected forest of Mount Yawila.

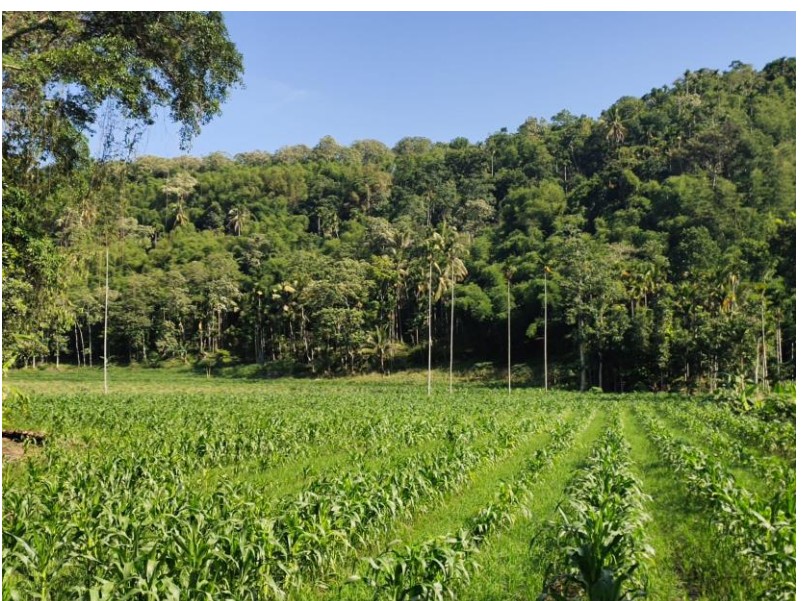

**Figure 2.** Indigenous *kaliwu* agroforestry system (IKAS) in Sumba.

The ethnic Sasak and Sumbawa people in the NTB province practice traditional agroforestry, including *rau*, *turi* fallow, hedgerows of leguminous trees, *kebon*, and forest gardens [24] (Figure 3). *Rau* is mainly found in dryland agricultural areas with scattered tree coverage. In some areas, small catchments of *rau* can be functional for rainwater storing systems [25,26]. The *turi* fallow system involves planting *turi* (*Sesbania grandiflora*) regularly in one row with a wide spacing between the plants and rows. Generally, the crops that are planted in the spaces between *turi* rows are cereal crops. The *turi* fallow system prevents erosion, maintains agricultural land fertility, and produces green manure, fodder, and fuelwood for households [27].

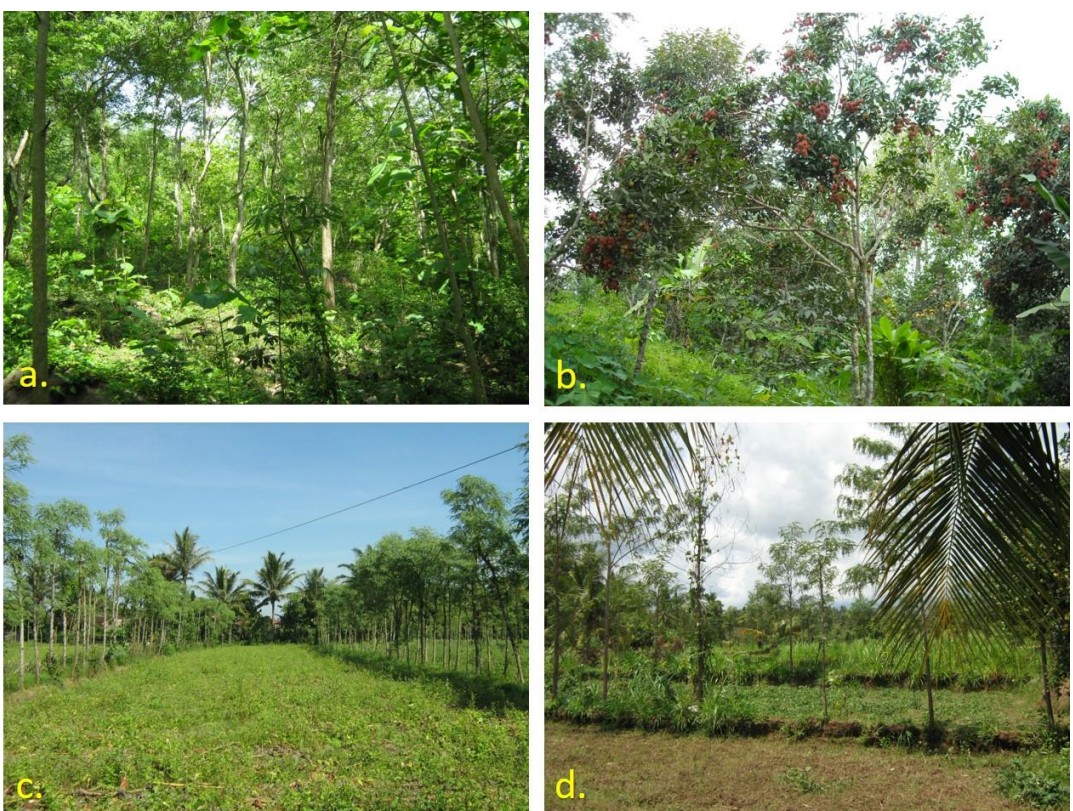

**Figure 3.** Agroforestry patterns in Sasak and Sumbawa: (**a**) *rau*, (**b**) *kebon*, (**c**) turi fallow, and (**d**) hedgerows with leguminous trees.

Similar to the *rau* system, the *kebon* system combines trees that produce fruit or timber with food-based crops such as cereals, vegetables, and legumes. *Kebon* systems are located near settlements. A forest garden is another type of agroforestry system in which the land is typically owned communally, but the trees can be owned individually for various purposes [24,28]. The ethnic Sasak and Sumbawa people have also adopted a land-capability- and location-based cultivation approach [29].

The ethnic Sasak and Sumbawa people recognize several land use arrangements that can be used for watershed management. The distance to a settlement determines the land use arrangement [29]. *Gawah* refers to upper watershed wooded areas. *Gawah* is usually not an open area and is primarily agricultural land with non-anthropogenic activities. In several places, sacred forests can be found that provide water-regulatory functions in a watershed. The ethnic Sasak and Sumbawa people have an area in which horticultural crops are cultivated called *kebon* that fulfills their food needs. *Kebon* also includes sites for growing other crops, such as tobacco, coffee, and other plantation products. In areas with a flat topography and on riverbanks, land use systems called *Bangket* can be found. *Bangket* systems are paddy fields located around settlements, where the harvesting frequency depends on the water availability and is mainly once a year due to the short rainy season period.

Regarding regulations, the ethnic Sasak and Sumbawa people have a system called *awig-awig*. Jayadi and Soemarno [30] explained that *awig-awig* is a type of local wisdom with customary rules (customary law). Mukhtar et al. [31] explained that *awig-awig* is a type of local wisdom owned by the community. It contains values or norms that evolve, beliefs that are expressed by myths and symbols, and certain symbols that blend with culture across generations [30].

*Awig-awig* also relates to managing natural resources in a watershed from a downstream to an upstream watershed. There are prohibitions for cutting down the trees of a

*gawah* (forest), especially in sacred forests. They believe it will bring calamity and disaster, such as floods, landslides, and pest attacks. Regarding water resource management, there is a type of local wisdom related to the management process (planning, implementing, and monitoring). At the planning level, the term *takepan* is known as a moral message and advice given by entertainment/singing, which usually brings up ideas from traditional advice and leaders to plan a form of cultural attraction to promote environmental awareness. *Nyampang* is known at the implementation level as the activity of planting trees and preserving nature, especially in the catchment area of water sources (e.g., springs). Another form of monitoring is widely known as *pamali* culture. In *pamali* culture, tree utilization in a forest is adjusted to the characteristics of the trees. Trees that produce fruit for consumption are not allowed to be cut down, while specific trees for timber production are allowed to be cut as long as they are replaced with new seeds.

The banu tradition of the Atoni (Atoin Meto) people on the island of Timor prohibits the use of natural resources for a certain period of time (based on a customary agreement) in order to provide opportunities for nature (flora and fauna and their ecosystems) to carry out ecological restoration or regeneration [32]. The *banu* tradition has two stages of implementation. The first is *saeba banu*, a ritual that stops all community utilization of protected natural resources for a certain time. The second ritual, *sanu banu* (the end of the protection period), marks the community's resumption of the use natural resources in each communally managed area. The *banu* tradition protects natural resources from the species to ecosystem level in terms of socio-cultural, economic, and spiritual aspects. The application of the *banu* tradition and the customary sanctions in each Atoni tribal community unit may vary according to the characteristics of the protected natural resources. The *banu* ceremony is accompanied by the slaughter of cattle (cows or pigs) as a binder to start and end the collective customary agreement. Customary regulations regarding the management of natural resources in forest areas and communal land are also found in several tribal communities on Sumba, Flores, Alor, Rote, and Sabu in the province of East Nusa Tenggara.

### 3.3.2. Java: A Densely Populated Area with High Rates of Population Growth and Land Conversion

The island of Java is the center of the government of Indonesia and is a symbol of modernization in Indonesia. Accessibility is already well available in almost all corners of the island, and its high population density and diversity of behavior has caused customary values to be rapidly eroded. However, until now, some traditional community groups have still cared for and implemented their TK in their daily life to protect the environment.

In some areas of Java, such as the Temanggung regency, the local community still practices the *nyadran kali* tradition [33]. The *nyadran kali* tradition thanks river waters by praying and cleaning the river. This tradition involves agricultural systems, especially irrigation, so farming continues during the dry season. The local community also plants bamboo around the river [33]. They believe that bamboo absorbs water well. Bamboo is very suitable for land rehabilitation and SWC, especially for soil erosion control [34]. They also practice the *ili-ili* tradition. An *ili-ili* is a person responsible for regulating the flow of water during the dry season. *Ili-ili* comes from fluent Javanese *mili*. *Ili-ili* is also a person who owns agricultural land in the area. An *ili-ili* must ensure that every agricultural plot is evenly irrigated. They work voluntarily and without pay for the common good of a bountiful harvest. In this tradition, everyone should participate in overcoming the irrigation problems occurring during the dry season in rainfed farming systems [33].

Regarding the TK of watershed management, the local communities in Java are familiar with *pranata mangsa*. *Pranata mangsa* was established by Sri Susuhunan Pakubuwono VII, the king of Surakarta in central Java in 1855, as a guide for understanding the seasons based on the solar system [35]. The practice of *pranata mangsa* advises humans to mitigate seasonal changes to minimize their environmental impact. Seasonal vulnerability during the dry season is already known to the community based on inherited knowledge, personal experience, and the observation of climate change factors. In the *mangsa katiga* (dry months),

there are early warnings or signs of natural dangers due to the dry season, which are indicated by falling leaves and reduced water sources. Meanwhile, the rainy season is marked by new shoots and leaves appearing on several types of plants, such as tamarind and yam. The landscape features determine the natural hazard, which can be derived from local knowledge depending on the season. Spatial or geographic contexts differentiate the application of *pranata mangsa* in terms of hazard and risk identification [36,37].

The Urug community, who live in the Bogor regency (west Java), uses traditional terraces for rice fields in mountainous areas, using water from the Urug river to irrigate their rice fields through *susukan* (traditional reservoirs) using ditches. There are people known as *ulu-ulu* who ensure that all the farmers receive water. The indigenous people of Urug also practice *gotong royong* (cooperation) among the community members who are involved in traditional agricultural irrigation [38]. In addition, the Using (or Osing) community in Banyuwangi regency has local knowledge regarding the use of water in irrigating paddy fields. Destroying forests in the spring areas is also prohibited. The arrangement of the water distribution and the use of irrigation canals is carried out by a *modin banyu*, a person appointed based on deliberations and farmers' agreements. The rice field irrigation system is channeled from major rivers and distributed to all the farmers who use water [39].

Other types of TK are found in the community in the Tawangmangu district, which until the early 1990s consumed corn and tubers as their staple foods due to the high rainfall and steep slopes in the area [40]. In general, there are three major themes related to the TK of the Tawangmangu community regarding non-rice food security. The first is the inheritance of traditional ecological knowledge (TEK) through stories about the origin of vegetables and corn, taboo words related to the prohibition of planting rice, and the meaning of symbolic processions in traditional ceremonies for cleaning the village and protecting water sources, as well as corn offerings for ancestors. The second is how their society views God, ancestral spirits, and other living things. For example, people must ask the tree spirits for banana leaves and must avoid killing animals such as ants. The last involves strategies and practices for conserving natural resources, including selecting non-rice food crop commodities, managing agricultural land (terracing and irrigation systems), cropping patterns (intercropping), managing water resources, and post-harvest management. Local communities also protect the forest areas on the slopes of Mount Lawu and Pringgodani Cave by banning logging and reforestation [41].

### 3.3.3. Sulawesi: Areas with Low Population Density and High Hydrometeorological Disaster Potential

The island of Sulawesi, located in the middle of the Indonesian archipelago, is inhabited by various tribes spread over five provinces, namely South Sulawesi, Southeast Sulawesi, West Sulawesi, Central Sulawesi, Gorontalo, and North Sulawesi. The various major tribes on this island include the Makassarese, Bugis, Toraja, Mandar, Buton, Tolaki, Muna, Pamona, Kaili, Banggai, Gorontalo, Minahasa, Talaud, Mongondow, and several other tribes. Although the people on this island have been infiltrated by cultures from outside, especially Java, Asia, and Europe, as well as by cultural exchanges within the island itself, the awareness and adherence to traditional values, norms, and beliefs in responding to the surrounding nature is still relatively high. Some communities have local wisdom in terms of regulating land use in order to maintain the land's carrying capacity and to keep it sustainable. They usually designate certain parts of the land as a forbidden forests and other parts as available for supporting for human life.

In the province of South Sulawesi, there are customary rules for the community who have settled at Lake Tempe, which regulate the location of fishing grounds, places of residence, areas where fishing is prohibited, and areas where vegetation grows floating in the water in Lake Tempe [42]. The local wisdom also applies land use regulations into several zones within their customary areas, such as the Karampuang community in the Sinjai regency [43] and the Cerekang indigenous community in the East Luwu regency [44].

The *kajang* community in the Bulukumba district maintains the forest environment by applying the values of *pasang ri kajang* as a form of local wisdom through the terminologies of *jagai linoa lollongbonena kammayya tompa* (protect the earth and its contents) and *kasipalli* (do not destroy the forest) [45]. Their local wisdom is manifested through a "sacred forest" in order to control forest resource utilization, which the local people obey due to their belief in punishment from nature if they carry out destruction [46]. This local wisdom is under the supervision of traditional leaders (*Ammatoa*) who have an exemplary and great influence on forest conservation in the communities around the forest. The *Ammatoa* exhibit the values of leadership, trustworthiness, and steadfastness and prioritize the *abborong* (advice) principles in their decision-making [45].

In the Toraja regency, a highland in northern South Sulawesi, the community has a form of traditional knowledge called *ma'pesung*. They believe that every spring becomes a place of worship, so the area and landscape around the spring must be maintained [47]. They also use sweet potato leaves as ground cover for erosion control and foraging for pigs [48]. This pattern has become the culture of the Toraja people, who have used pigs as an important part of their traditional rituals since the time of their ancestors. This plant is combined with rows of pineapple plants to strengthen terraces under teak trees (*Tectona grandis*), especially in sloping areas, as seen in Figure 4. The fast-growing sweet potato leaf has a bowl-like shape that dampens the kinetic energy of rain, and its stems and roots that spread over the ground can protect soil from erosion. The harvesting model with a rotation system that they use allows the surface soil cover to last throughout the year.

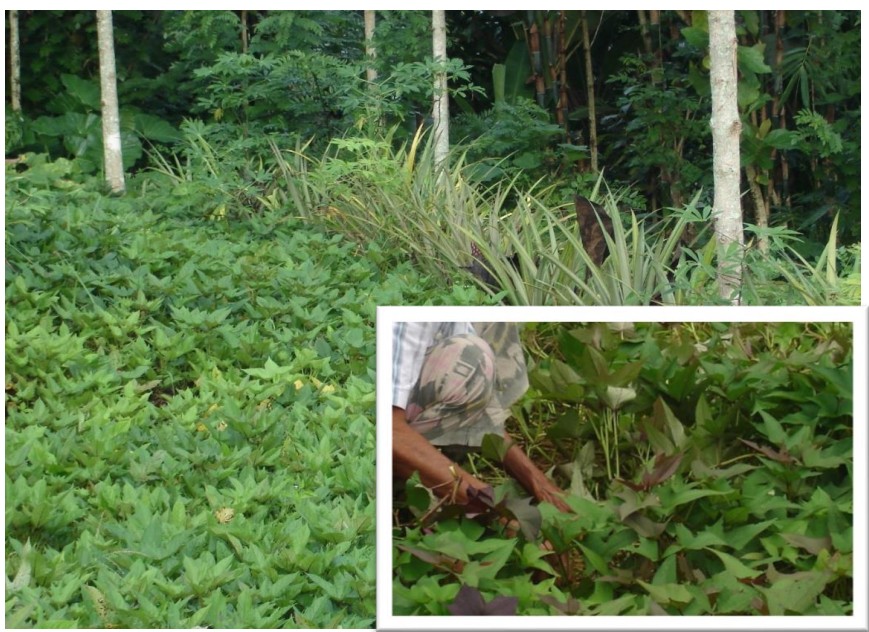

**Figure 4.** Sweet potato plants planted between rows of pineapple plants and teak trees (*Tectona grandis*) in Toraja regency.

For the Toraja people, paddy fields, gardens, and forests are sacred and should be used, protected, and maintained properly. The management of this heritage system is the responsibility of the *tongkonan* (traditional house), which also serves as the center of life management for the Toraja people (Figure 5). In the Torajan system, plots of land commonly planted with bamboo, mountain pine (*Casuarina junghuhniana*), and cempaka (*Ermerilla ovalis*) as *tongkonan* building materials or traditional ceremonial huts can be passed on to posterity after going through certain traditional ceremonies [49]. The community plants their customary lands with bamboo and trees to fulfill the raw material requirements for housing and ceremonies. This makes the community forest development activities successful in Tana Toraja.

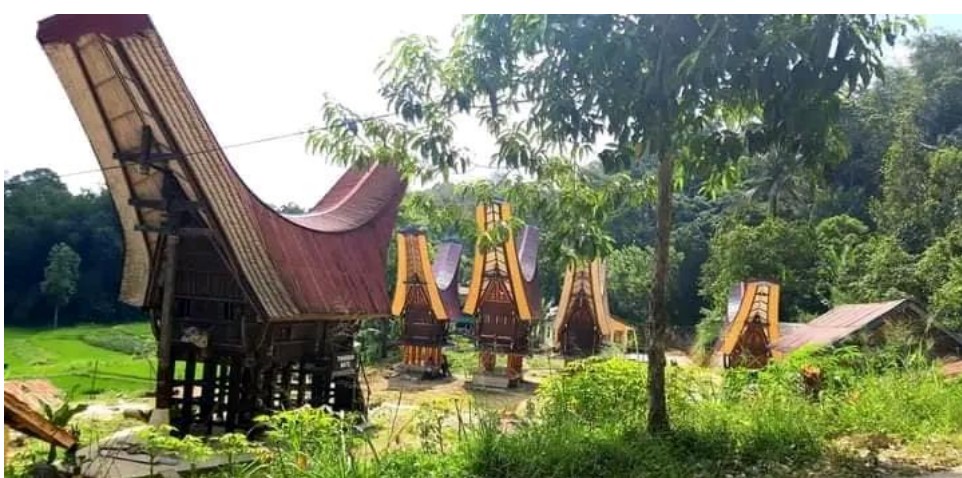

**Figure 5.** *Tongkonan* as a center for the activities of the Toraja indigenous people surrounded by community forests.

However, in line with the times and the increase in the population, land conflicts have become the social problems emerging in the Toraja and north Toraja regencies. For example, in the Mangkendek sub-district, the community has begun to view the *tongkonan* land from an economic perspective, which was triggered by the government's plan to build an airport in that location. The community competes for compensation for tongkonan land ownership [50]. However, the resolution of problems in the Torajan society is generally resolved by local customs. Legal action is taken if the conflicting parties do not find a middle ground. At the court level, a *to parenge* and other traditional leaders resolve problems at the customary level by giving recommendations. A *to parenge* is a person elected in a traditional meeting and has responsibilities in society, especially in regulating the order of social life. A *to parenge* does not carry out their duties alone but in collaboration with traditional institutions consisting of three people [50].

The people of the Minahasa district in North Sulawesi Province have a TK practice called *mapalus*. This traditional philosophy was born, grown, and developed from community awareness always growing together. The practice of *mapalus* is a form of monetary empowerment with the idea of developing collectively and is primarily based totally on TK. It requires its members to be independently wealthy and capable of enhancing the welfare of others [51]. In addition to the wisdom of mapalus, in the Minahasa regency, there is a saying "*si tou ti mou tumou tou*" (people have to live by supporting others). The Minahasa community has trusted these two things for generations to improve the welfare of each of its citizens.

Other forms of traditional community knowledge found in Sulawesi in the SWC context include the practice of *ilengi* in the Gorontalo province. *Ilengi* is an agroforestry system that forms a stratified plant canopy that reduces the speed and size of raindrops to minimize their kinetic energy when hitting the soil, thereby increasing the effectiveness of erosion control [47].

Local wisdom regulating land use and ownership in preserving natural resources and the environment can also be found in Southeast Sulawesi Province, namely in the Buton community [11,52]. Furthermore, in Central Sulawesi Province, the Kaili tribe also has a set of local knowledge customs that is manifested in their daily lives, such as forest conservation, Lindu lake waters, taboos in speech, and other traditional ceremonies. The harmony of life described from the local wisdom of the Kaili people always balances the bond between man and nature [53].

### 3.3.4. Kalimantan: An Area with Extensive Tropical Forests Rich in Minerals with High Potential for Forest Fires

Kalimantan, also known as Borneo, is inhabited by four dominant tribes, namely the Dayak, Kutai, Banjar, and Melayu tribes. Of the four tribes, the tribe that is often considered to still live in harmony with the forest and the environment is the Dayak tribe. Dayak livelihoods are heavily dependent on forests and other natural resources [54–56]. They believe that land and society are interdependent. The term *latitana*, or forest land, is a basic philosophy regarding land use in various aspects of life, so preserving natural resources becomes a vital necessity [57].

Some scientific articles also mention them as being a forest community [58] due to their economic activities that are related to the forest being carried out only at certain times, such as collecting various non-timber forest products (NTFPs), i.e., agarwood, rattan, and medicinal plants, as well as fishing and hunting [56,59]. The term tradition has the meaning of regulation in the Dayak Benuaq language. Their traditions have been inherited for generations and accepted as customary law that binds the activities within the Dayak community [56]. Upland rice farming traditions influence the traditional forest management systems, upland land tenure, prohibitions, and regulations in terms of land and other land uses and natural resource disputes [55]. The Benuaq people in East Kalimantan Province prohibit forest areas from being used as agricultural fields [55]. Consequently, strict social and spiritual sanctions apply to violators of this rule.

They have developed a unique agricultural system characterized by a mosaic of forest regrowth [55,56]. The Tunjung and Benuaq groups in Barong Tongkok, East Kalimantan Province, use the practices of *munaant*, *simpukng*, and *lembo* as yard management practices [55]. *Simpukng*—the indigenous forest garden system—is one of the forms of TK used in their agricultural intensification systems combined with their forest domestication practices that provide economic value for the Dayak people. Through the use of *simpukng*, they grow various fruit plants with different fruiting seasons, such as rattan, bamboo, wood, and other plants, so it is possible to harvest them throughout the year [57]. A study on the Tunjung and Benuaq groups reported that they grow about 91 useful plants in their forest gardens and cultivate dozens of local rice varieties in their upland rice fields [55]. Furthermore, they also regulate the use and inheritance of *simpukng* in the form of complex customary law to avoid over-exploitation [57].

Along with their nomadic lifestyle, the Dayaknese people practice swidden agriculture that begins with land clearing by burning. Then, they grow paddies in unirrigated rice fields in the highlands (in the Dayak language this is called *umaq*), which are also planted with vegetables, bananas, cassava, pineapples, and other crops. This is a traditional agricultural system that involves complex rotations of each crop and tree species in a specific land unit [57]. They abandon this land after several years of cultivation so that the soil's fertility recovers naturally. The Dayaknese people monitor each other when burning land. They recognize this tradition as *handep* or *hapakat*, which means cooperation in the Dayak language [60]. The practices of *handep* and *hapakat* are relatively easy to achieve because, traditionally, groups of Dayak people live under the same roof in one longhouse, also known as a *betang* house. Simple fire extinguishing is carried out by using tree branches. Along with the increase in the population and the expansion of land ownership, the practices of *handep* and *hapakat* are challenging to carry out because they cannot extinguish fires on a larger scale.

Collecting NTFPs is one of the practices for utilizing forests in a sustainable manner. Gaharu—the valuable aromatic resin formation of agarwood—is an NTFP usually collected by the Penan Benalui and the Kenyah Badeng people in East Kalimantan Province [59]. They apply local knowledge to assist them during collection expeditions, such as species differentiation, spatial distribution patterns, plant associations, the morphology of infected trees, and regeneration [59]. During their collection expedition, they utilize the practice of *sekau* rather than gaharu. They consider it potentially harmful and unfortunate unless it is mentioned among dealers and non-locals [59]. On the other hand, the Dayak Benuaq

people mostly extract rattan from their forest gardens [55,57]. The difference between the NTFP used by the Dayaknese people highly depends on the natural resources around them.

*3.4. Limitations and Constraints*

Based on their characteristics, forms of TK have their limitations and constraints. The existing TK cannot overcome all the problems of poverty and the multi-dimensional biophysical and socio-economic problems regarding watersheds [61]. TK is generally pragmatic and temporary and is characterized by knowledge that is firmly rooted in local conditions and is specific, so it generally requires adjustments and cannot be applied in other regions [26].

Local wisdom is also easily eroded by broader economic and social forces. Interactions with more advanced outside communities often form new social structures that can undermine customary values. The socio-political and cultural influence of the West, which tends to dominate global affairs, is a serious challenge to indigenous knowledge existence [62]. National and international economic growth, the implementation of homogenous local government and education systems, and the intrusion of consumptive trends and culture increasingly leads to the homogenization of cultures that erode beliefs, values, customs, knowledge, and customary practices that have previously been the basis of indigenous people's lives. Lwoga et al. [63], in their study in Africa, found that farmers regard TK as an outdated knowledge system. The same occurs with young people. They find it difficult to accept the transfer of TK due to modernization and formal education systems, which lack any recognition of TK.

From their research on the Dayak Paser indigenous community in East Kalimantan, Nugroho et al. [64] found that there was a change in the behavior of the indigenous people living around the Gunung Lumut protected forest, who had been known to hold and apply the norms, beliefs, and traditional wisdom and knowledge of their ancestors that was due to an increased desire for modern life, better accessibility, and socio-cultural assimilation with immigrant communities. Nugroho et al. [64] believe that without efforts to maintain traditional values, a strong relationship with nature, which is often important to the identity of indigenous peoples, will not be able to last much longer. This is based on the real conditions on the ground, where even remote communities are forced by the 'needs of life' to convert their ancestral forests into rubber and oil palm plantations in order to simply survive.

Among the Dayaknese people, TK is not something common. This is because they limit the transfer of knowledge to the family line, usually from parents to sons [54,58]. As happened to the Dayak people in the Mempawah, Sanggau, and Landak districts in West Kalimantan Province, younger generations are less familiar with various plants and their health benefits than older ones [54]. If this continues, TK will be lost before scientists and authorities can identify it and transmit it as a solution to reduce the rate of land degradation. Traditionally, the Dayaknese people view the use of *simpukng* as common property rather than private property [57]. On the other hand, the unclear ownership status coupled with the existence of a forest management permit from the government for the private sector without the consent of the local community potentially becomes a threat to natural resource reservations. Research in West Kalimantan Province has stated that the Dayaknese people in Jagoi Babang village have begun to abandon their local wisdom of swidden agriculture [65]. It has been further stated that in the Mawas forest area, Central Kalimantan Province, land clearing by burning for swidden agriculture has been uncontrolled [60]. This can trigger forest and land fires [66].

The government's next priority should be to not only protect and provide space for indigenous peoples politically and to ensure their inherent rights (including customary forest areas) but also to preserve the TK that has existed so far so that it does not disappear by itself because it cannot be preserved by indigenous peoples.

## 4. Sociotechnical Measures: Theoretical Frameworks

Facing worsening environmental problems such as the loss of biodiversity and resource depletion, a fundamental change in the system is needed, which can be briefly conceptualized as a socio-technical approach. A socio-technical system connects the two

parts of a water management system, one being the infrastructure and the second being the human system and the interaction between the two [67]. Involving a complex interaction between humans, machines, and the environmental aspects of the work system [68], this approach can solve natural resource management problems [69]. However, the fulfillment of social functions involves not only technology but also culture, public policies, business models, markets, and infrastructure [70].

In addition to solving watershed management problems, a socio-technical framework can also be used as an instrument for approaching the design of watershed management, both in terms of technical and socio-economic aspects as well as institutional aspects. A socio-technical system is a system of interaction between the social environment and technology in a coherent and interactive manner, where human activities are facilitated by using the technology and vice versa. The intrusion of a technology is facilitated by social, economic, and institutional acceptance [71,72]. The better a social system is integrated with a technological system, the better the results that will be achieved [73]. A socio-technical system consists of social and technical components that contribute directly or through other components to the common goals of the system [74]. Clegg et al. [75] described the socio-technical approach as a framework of complex relationships between technical factors (technology, infrastructure, processes) and social factors (people, culture, goals), where changes in one element in the system will cause/require changes in the other elements.

Table 1 presents examples of fundamental problems regarding the five aspects that are commonly found in the upstream watersheds based on the research experience of the authors.

**Table 1.** Social and technical problems in watershed management.

| Aspect | Aspects Related Problems | Underlying Issues |
|---|---|---|
| Goals | - Market unavailability<br>- Unavailability of fertilizer and good-quality seeds | - Limited accessibility |
| Culture | - Swidden farming<br>- High dependence on natural resources | - Minimum farming capital<br>- Lack of simple and efficient technology |
| People | - Minimum population at the productive age<br>- Low human resource capacity/knowledge/skill | - Urbanization of productive age to urban areas for better livelihood<br>- Limited capital and equipment |
| Technology | - Subsistence land management system<br>- No product preservation technology | - Lack of capital<br>- High attachment to traditional values that are not compatible with modern technology |
| Infrastructure | - Minimum accessibility<br>- No warehouse for agriculture product storage | - Lack of capital<br>- Minimalist/subsistence culture of life |
| Process | - Controlled by a middleman<br>- Direct selling system/no product processing | - Low human resource capacity/knowledge/skill |

Figure 6 gives an example of an approach to problem solving in watershed management using a socio-technical framework that was developed based on the experience of the authors in the field.

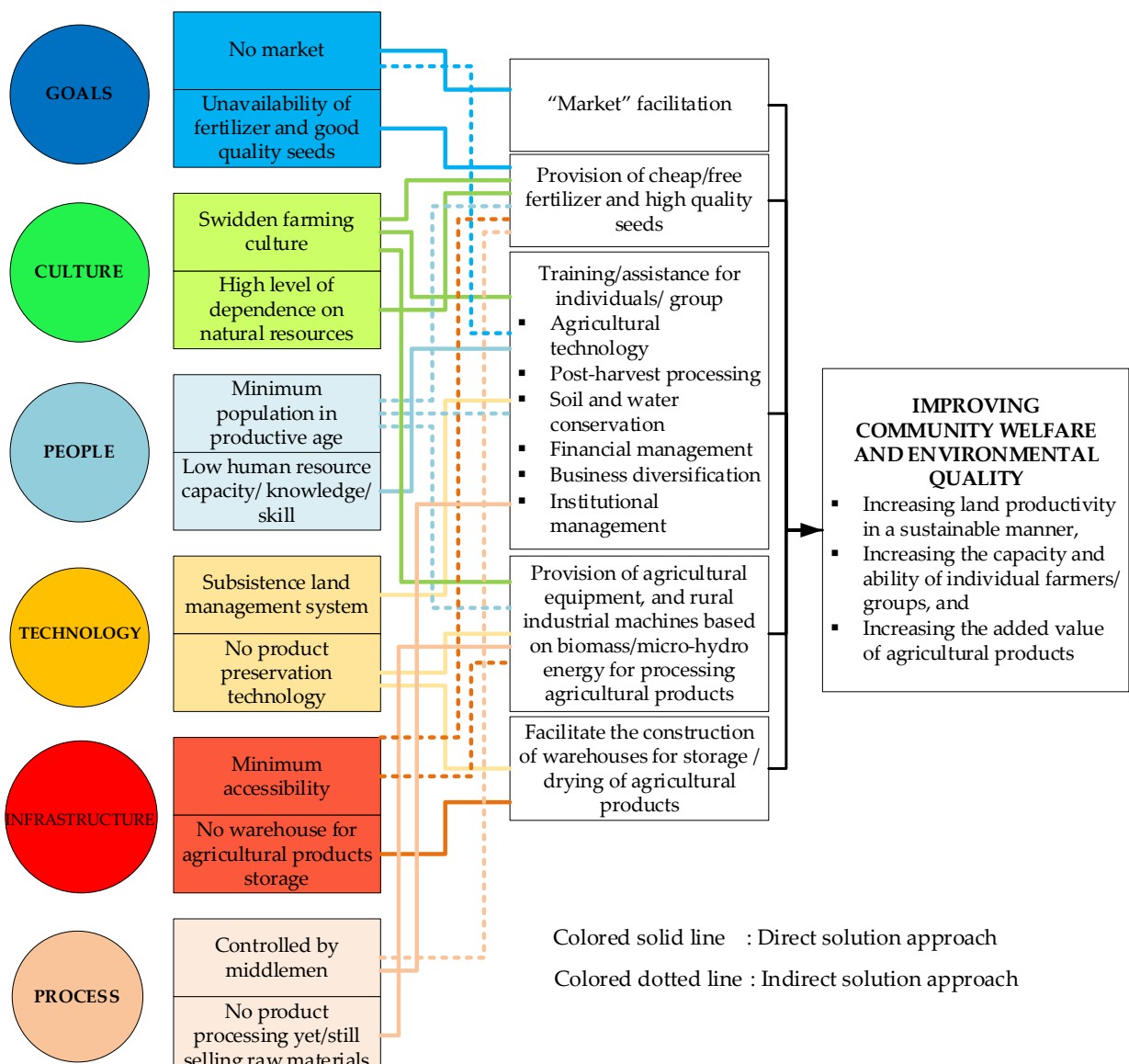

**Figure 6.** The sociotechnical framework in the problem-solving formulation.

## 5. Incorporating Traditional Knowledge into Science-Based Sociotechnical Measures

The development of the global population and the impact of climate change has raised the challenge of a need for environmental governance based on new knowledge to better address complex socio-ecological processes. TK and the modern scientific knowledge of nature has become the center of attention, including efforts to integrate them both into natural resource management policies by combining all experiences, skills, and lessons learned from empirical scientific evidence to achieve a fair and sustainable use of natural resources. Unfortunately, the existence of TK and local wisdom is often displaced by advances in science and technology. Technology and local knowledge can be synergized to become a great force in supporting the management of natural resources that are vulnerable to degradation, such as those in upstream watersheds. On the other hand, scientific knowledge is built based on systematic quantitative technology and methodology, but it is generally weak in terms of long-term experience, which is the main characteristic of

TK [64]. In the future, it will be necessary to incorporate TK into sociotechnical measures that are based on science.

The synergy of these two potential systems can be incorporated into the management stages of watershed management, starting with planning, implementation, and monitoring and evaluation (M&E) activities. Activating stakeholders, creating networks, preserving trust and commitment, mapping watershed conditions based on local knowledge using the most recent technology, determining land use directions based on participatory mapping involving local communities and all stakeholders, and being integrated with the newest mapping application tools such as GIS, are some examples of how these terms can be used [76–80].

## 5.1. Planning Stages

At the planning level, knowledge integration involves exchanging and internalizing information from scientific knowledge with local community knowledge that is embedded in the behavior, values, norms, and culture of local people [81]. However, socio-technical systems are not only related to certain actors, institutions, and physical characteristics but also involve practices and procedures, protocol frameworks, and the wider context in which the technical approaches and social aspects can be integrated. In this context, in terms of solving problems, it is not only data but also the types of data and the procedures for obtaining the data and information that is important, which are determined through a social approach and not just through technical means [82]. Different types of land use managed by different communities have different ecological, economic, and socio-cultural impacts [83–86].

Setiawan and Nandini [87], in their research regarding the process of prioritizing sub-watersheds using the integration of principal component analysis (PCA) and a weighted sum approach (WSA), suggested considering socio-economic parameters in the prioritizing process, thus combining socio-economic and cultural factors with biophysical factors for the prioritization of results in a more reliable model and with a broader intuitive perspective.

Figure 7 presents a socio-technical approach framework for watershed management technology planning formulated based on the results of the author's study. This planning framework involves a spatial modeling approach within a DSS (decision support system) framework and a demonstration plot approach. The development of the demonstration plots involves the community and considers the technical feasibility, financing potential, and suitability of the system with regards to socio-cultural and community resource capacities. Through the utilization of spatial models verified by field trials following the community's specific character, this technological approach is believed to be effective in solving the problems of watersheds. All processes are carried out in a participatory manner involving key stakeholders, starting with goal formulation, followed by policy formulation through an interactive and adaptive process [88].

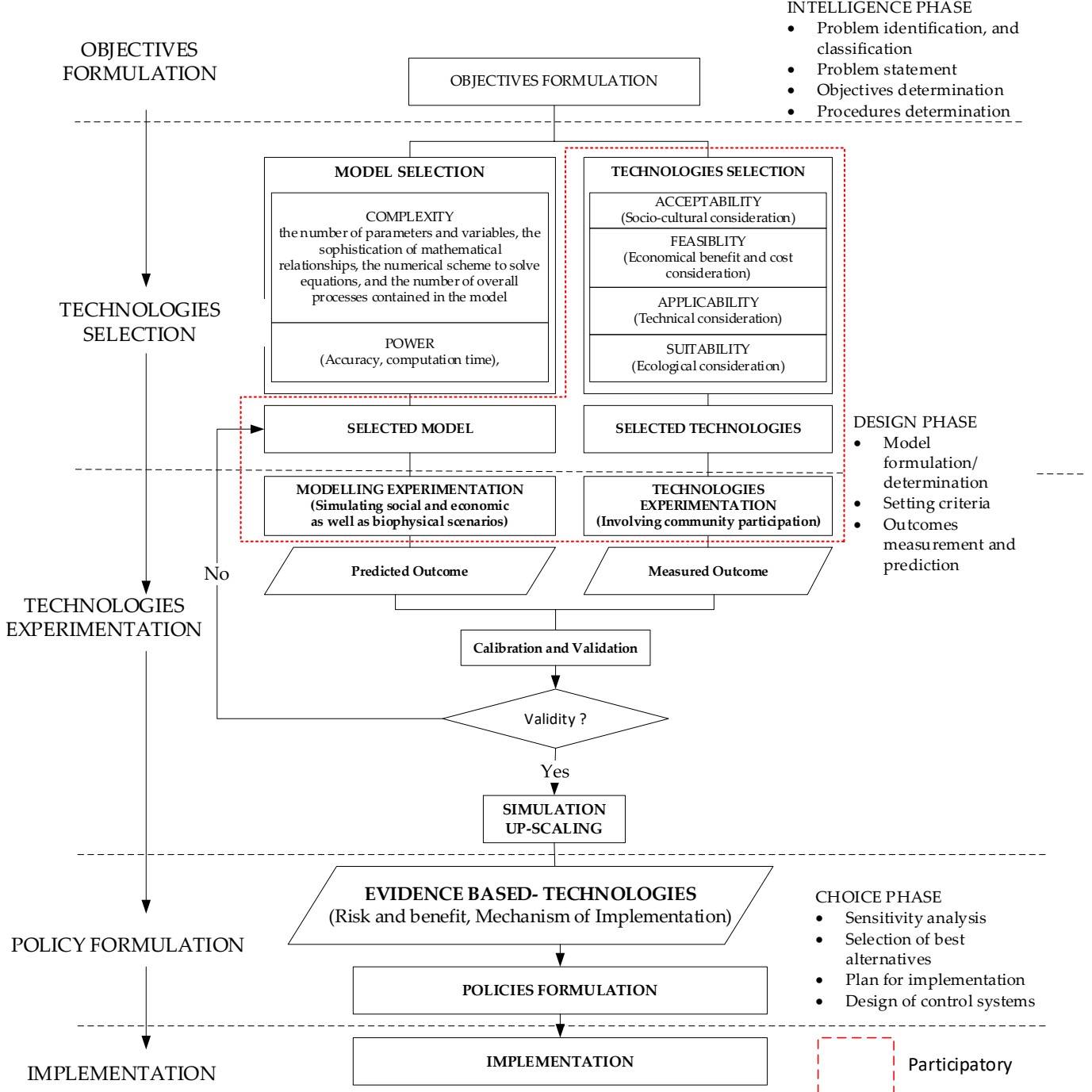

**Figure 7.** Socio-technical system framework for watershed management technology planning, adapted from Nugroho et al. [89].

A detailed framework for designing land rehabilitation and soil and water conservation techniques using a socio-technical approach is presented in Figure 8. In practice, selecting appropriate soil conservation techniques is carried out by considering various factors, including biophysical, social, economic, and cultural factors. The technique being applied must be biophysically suitable, economically feasible in terms of implementing it with local resources as well as being profitable, socio-culturally acceptable in accordance with the community's conditions, and institutionally applicable in terms of increasing land productivity without causing damage to the environment. The process of formulating

technology for land rehabilitation and soil and water conservation (LR&SWC) using a socio-technical approach is described in the following flow chart.

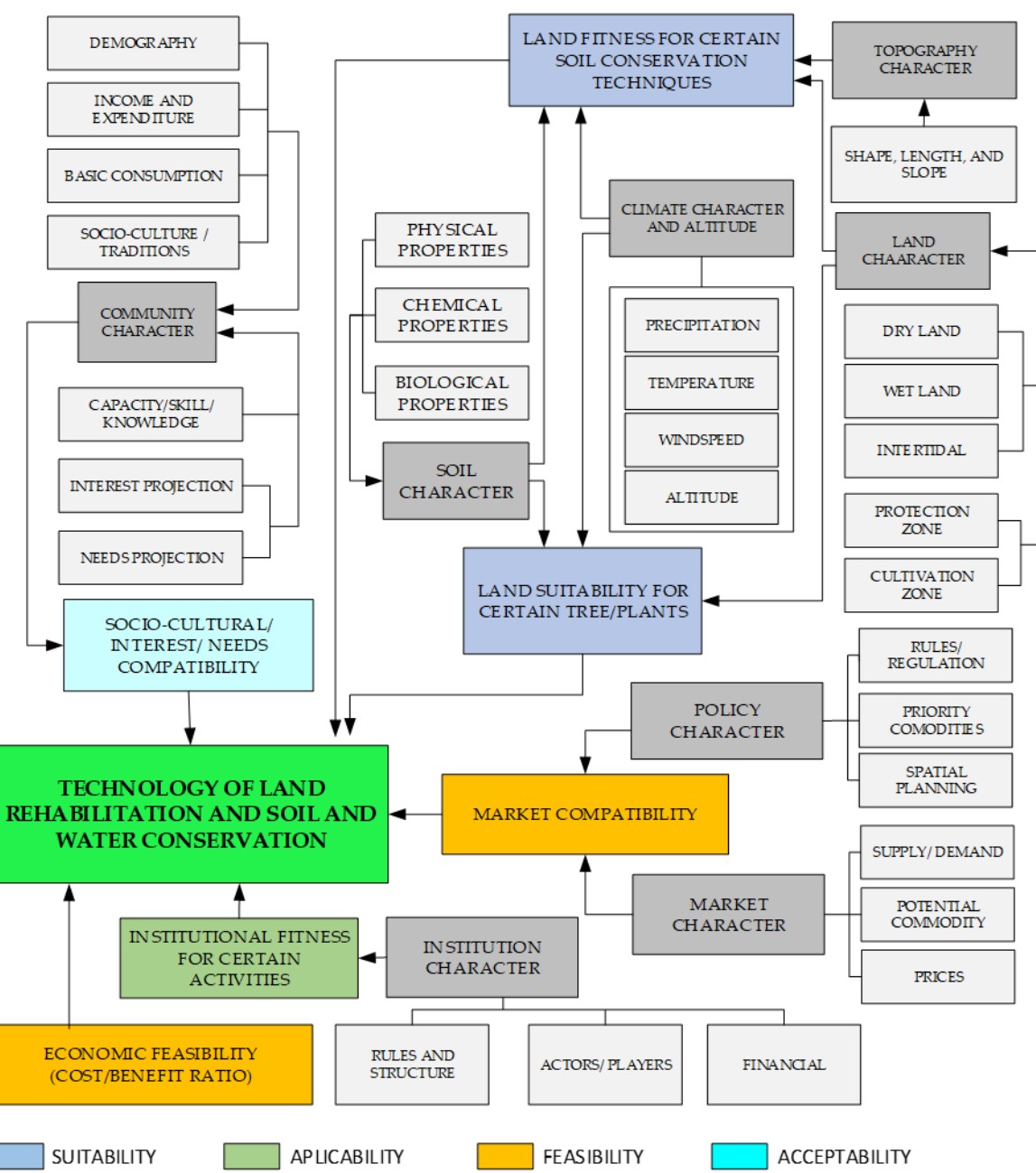

**Figure 8.** The use of a socio-technical approach in formulating LR&SWC technology.

However, socio-technical systems are not only concerned with people, institutions, and technical instruments but also include procedures, protocols, and modes of delivery that are acceptable in the wider context of the process and stakeholders [82].

## 5.2. Implementation Stages

The involvement of various stakeholders is an important factor in watershed management in an integrated watershed management framework, including the community residing in the watershed. Communities with already-acquired local knowledge have the ability to overcome the problems related to watershed management, such as maintaining adequate land cover, water use regulation, the construction of soil and water conservation, and river utilization and protection regulations. The implementation of local wisdom has

shown great potential in contributing to ecological conservation [90]. Local knowledge and wisdom provide a better understanding of flood vulnerability and can provide information to decision-makers and stakeholders to determine adaptive measures that can be applied to increase the community's resilience to flood disasters [79]. Integrating the local knowledge systems practiced by the community into modern knowledge can reduce the risk of flood disasters [91]. The interaction and integration of local and modern knowledge encourage the process of knowledge construction toward a more accepted understanding of modern science [92]. However, in its implementation, each community has a different method according to the conditions of the area where they live. In integrating local and scientific knowledge, no single approach can stand alone. The integration process needs to be carried out systematically, reflexively, and continuously involving several views and methods related to environmental management issues [93].

The following form of incorporating TK into science-based sociotechnical measures at a more advanced level is carried out by community groups and involves one or two other institutions to jointly manage watersheds. This watershed management network is a little more advanced because it engages various parties outside the local community. As stated by Indrawati [94], cooperation networks in the management of upstream watersheds can be complex because they involve more stakeholders, namely the people as the landowners, local governments, forestry agencies, agricultural agencies, and agencies related to rivers. One example of upstream watershed management that has buildt a simple network can be found in the village of Tawang Sari, Brantas sub-watershed, east Java, which involved Perum Perhutani as concession owner and community living around the forest area in forest management [95]. The same case was found in the village of Suka Maju, Air Nipis district, South Bengkulu regency. In this village, the watershed management was carried out by combining the community's knowledge of ecology with the modern management practices of community groups from nearby villages. The network built for watershed management was more expansive because it involved NGOs and local mass media to promote the various forms of outcomes or benefits that the communities received from the watershed management [96].

At a more advanced level, incorporating TK into science-based sociotechnical measures not only involves community groups or building networks but also utilizes various communication applications in order to study, practice, disseminate, and simultaneously maintain the sustainability of the existence of local wisdom in the community in terms of watershed management. One example can be seen in the community of the village of Beruk, which has local wisdom in terms of cooperation in order to maintain and prevent forest damage and minimize disasters. The people of the village of Beruk also have regulations that prohibit cutting down trees in springs, steep areas, and near waterfalls, as well as preventing erosion by piling stones on sloping land, which is better known as *nyabuk gunung*. This local knowledge and wisdom have been combined with scientific advances using the latest technology such as mobile phones and various communication applications such as WhatsApp [97]. Various communication applications, such as WhatsApp, have been used to form multiple WhatsApp groups in order to assist the public in disseminating different types of information more quickly.

One technological development milestone that changed how humans manage land was the emergence of the green revolution, which was namely a fundamental change in the use of agricultural cultivation technologies that began in the 1950s and continued to the 1980s in many developing countries. This green revolution changed the face of global agriculture, including in Indonesia. Agricultural production has increased significantly thanks to this revolution [98]. The phenomenon of the green revolution began to be applied in Indonesia, starting with the five farming business programs (*panca usaha tani*) followed by the seven business programs (*sapta usaha tani*), through which Indonesia achieved rice self-sufficiency. The green revolution (GR) period was a milestone for increasing global food security through a combination of mechanization and the widespread application of synthetic fertilizers, pesticides, and genetic varieties of high-yield crops [99], which

resulted in farmers depending on the use of artificial fertilizers, anti-pesticides, and superior seeds [100]. Research results in China have shown that a reliance on chemical fertilizers, pesticides, machinery, and energy use contributes to pollution and GHG emissions. The excessive application of fertilizers affects the quality of surface water [101–104]. The use of agricultural equipment mechanically affects the pressure on the land, which ultimately affects production [105].

This situation, among others, must be controlled by returning to an environmentally friendly land management approach. Local wisdom in terms of managing land by considering cultivation techniques, plant arrangement, and local traditional regulations such as those carried out by the Sasak [24,27,29] and Dayak tribes [57], for example, can be applied to improve and maintain the ecological balance of the environment. Returning to an organic farming system by utilizing existing local sources at a low cost is an alternative to sustainability. Organic farming, also known as agro-ecological farming, uses internal inputs such as pesticides and organic fertilizers, manure, compost, crop rotation, and biological pest control [106,107]. Cover crops reduce sediment loss and total P loss when sediment loss is high. Cover crops increase the dissolved reactive P loss in years with little sediment loss. One of the best management practices to reduce P loss is to place P fertilizers below the surface. Using cover crops is a site-specific management tool for reducing the sediment and total P loss [108].

However, this does not mean that non-organic agriculture should be abandoned or that organic farming should be solely relied upon. In situations with a high level of food demand, such as in Indonesia with its population of around 270 million people, the application of organic systems can be carried out simultaneously with improvements to non-organic farming systems. Organic farming can reduce global food insecurity, but this system cannot be the sole mainstay due to its low productivity. It is necessary to find the optimum point where combining inorganic and organic farming options can produce the best results [109].

In regulating the adequacy of plant needs, the local knowledge of *pranata mangsa* [36,37] can be applied and adapted to current conditions that are experiencing climate change. The presence of El Niño and La Niña affects the rainy and dry seasons in Indonesia. The efficient use of water is critical. Food security cannot only be achieved by increasing agricultural water sources, for example, through the construction of large-scale irrigation canals; it must be accompanied by increasing agricultural water use efficiency [110].

From the results of their research on indigenous peoples living around the Gunung Lumut protection forest, East Kalimantan Province, Nugroho et al. [64] stated that a holistic approach involving indigenous peoples, the recognition of customary law, and the revitalization of TK in line with the implementation of a scientific approach based on modern knowledge is a crucial step to ensure a prosperous society and sustainable forests.

### *5.3. Monitoring and Evaluation Stages*

The monitoring and evaluation (M&E) of watershed management are often carried out to ensure synchronization between the goals and achievements of watershed management, including the monitoring of socio-economic evaluations and community institutions as well as the biophysical performance of watersheds. M&E practices are based on the social dynamics of the community and the biophysical conditions in the watershed where there is a mutually influencing relationship. Community involvement in watershed management monitoring and evaluation activities effectively minimizes conflicts over using natural resources in a watershed [111,112]. However, there should be assistance and training for the communities in terms of developing programs and establishing partnerships [113].

Examples of incorporating TK into science-based sociotechnical measures at the monitoring and evaluation level can be found among people living on the slopes of Mount Merapi, central Java. They use social media, such as Facebook, and Facebook groups, such as the Merapi info group, which was initially intended to provide information in the context of monitoring and updating the eruption of mount Merapi, one of the most

active volcanoes in central Java, Indonesia, but has over time been used to provide other information such as the beauty of Merapi, ways of living in harmony with nature, and examples of various local wisdom practices and beliefs that exist in the community around Merapi [114]. Furthermore, monitoring and evaluating the implementation of TK can be conducted by utilizing global information system (GIS) tools. The monitoring of land use changes using GIS tools has found that applying TK in agriculture positively impacts efforts to reduce environmental damage [115].

## 6. The Way Forward: Ensuring Traditional Knowledge Sustainability

### 6.1. Mainstreaming Traditional Values and Empowering the Community

The sustainable management of watersheds is possible if there is an integration between the optimization of natural resources, institutions, technology, and sustainable funding and coordination, integration, synchronization, and synergy between programs and actors, including the local people living in the watershed. Community participation and mainstreaming TK and wisdom must be accommodated in watershed management, from planning to interactive monitoring and evaluation [116] in order to ensure long-term watershed management sustainability [117].

As social and spatial conflicts that lead to watershed damage have become a latent problem, holistic knowledge of the socio-economic and cultural dynamics of the community combined with a thorough understanding of the ecological system has become essential [89]. It is vital to seek and encourage the involvement of local people when adapting technology to their local conditions. This requires interactive participation between professionals and local people in order to bond them. Participatory technology development is a process by which TK and the research capacity of the community are combined with scientific institutions while strengthening local capacities to experiment and innovate. This increases the understanding of how a good watershed environment can impact communities and guides external agencies on how this engagement process should be carried out and how they should behave and accept traditional ecological knowledge as a valuable tool in developing alternatives to a solution [118]. In practice, participation means including communities more broadly and providing communities with balanced and objective information, consulting them for feedback and alternative solutions, and empowering communities by giving them full control over the key decisions that affect their wellbeing [119]. To ensure the integration of modern practice and TK, mobilizing indigenous peoples to participate in real programmed science-based activities is fundamental [118]. Traditional peoples and scientists can take steps to build strong foundations in order to anticipate future extreme climate change, such as increasing the resilience of public infrastructure and natural habitats [118]. Akbar et al. [120] stated that accommodating and integrating all the knowledge of the existing stakeholders could be a solution to improve the quality of the musrenbang (development planning deliberation) process.

Financial and technical capacities are essential factors in determining the ability of a community to participate in government programs [121–124]. Poverty reduces people's ability to use resources sustainably and increases environmental pressure. To survive, the poor and hungry make short-sighted decisions that can destroy their immediate environment [123]. Poverty is an obstacle to community participation in environmental development [122].

Local people should have greater access to information about the effects of these man-made situations so that they can adapt their TK, preparedness, and response patterns in order to reduce the risk of a disaster [125]. Various pieces of the scientific literature mention that leadership, social support, engagement, skills, and access to resources are important in empowering and building coalitions at the local level [126].

We suggest that TK, values, norms, and wisdom must be revitalized and empowered in line with formal law enforcement. Traditional wisdom, values, and norms are important risk-reduction instruments that should be valued and disseminated and should be incorporated into national and international disaster risk reduction (DRR) strategies [125]. Hartanto et al. [127] found that by triggering dynamic adaptation at the local level, state intervention

will strengthen customary institutions and their authority over natural resources. There is significant evidence that shows that economic development in harmony with local culture is more effective in the long term [128]. Fleming concluded that a successful indigenous economy is not a matter of which economies to support but how to support them [128].

Indonesia is very rich in its culture, noble values, and TK, which are the original resources of Indonesian society. A legal framework to protect these resources is urgently needed. Various parties, such as the government, research institutions, and academics, including the private sector, which has so far been associated with indigenous peoples, must establish effective collaboration to save Indonesia's original resources.

*6.2. The Documentation, Patenting, and Transferring of Traditional Knowledge to the Next Generation*

TK documentation is a formal way of ensuring that indigenous peoples' and traditional communities' social, cultural, and economic interests and rights are protected. TK documentation is basically the process of identifying, collecting, organizing, recording, and registering TK as a means of disseminating and protecting TK under certain goals [129]. These efforts can preserve and prevent further losses of TK and protect TK from unwanted use and claims by unauthorized persons [129]. When TK is in the public domain, commercial interests that can exploit it the most efficiently tend to benefit the most [130]. Indigenous people, as the owners, will lose the potential for compensation for utilizing their TK. However, TK documentation alone does not guarantee legal protection for TK. The protection of public knowledge through recognition of it as intellectual property rights (IPR) and its preservation by direct transfer from generation to generation through formal education channels, especially in indigenous territories and their surroundings, is urgently needed.

The Indonesian government has explicitly given recognition to indigenous peoples and their rights through the second amendment to the 1945 constitution, which was enacted on 8 August 2020, namely in Article 18b, Paragraph 2, and Article 28i, Paragraph 3. Article 18b, Paragraph 2 states that the State recognizes and respects the customary laws of community units and their traditional rights as long as they are still alive and in accordance with community development and the principles of the Unitary State of the Republic of Indonesia. Furthermore, Article 28i, Paragraph 3 states that cultural identity and traditional community rights are to be respected in accordance with the times and civilization.

Unfortunately, until now, there has been no law that specifically regulates indigenous peoples. The Draft Law on Indigenous Peoples, which has been discussed since the 2014–2019 period and was approved by the plenary meeting of the DPR Legislative Body on 4 September 2020, has not been ratified in DPR plenary meetings to date [131]. It is very urgent for the Indigenous Law Community Bill to be passed immediately in order to preserve culture, customs, TK, and customary land rights after the emergence of Law No. 11 from 2020 concerning job creation. This could become a formal legal basis for recognizing indigenous peoples' TK. Existing regulations that recognize the existence of indigenous peoples have indicated conflicts with investment, especially after the Job Creation Law was enacted.

The protection of TK is aimed at preventing unauthorized exploitation and maintaining economic and moral rights for holders of TK. In addition to having cultural and social value, TK has economic value. Without protection, there could be fraudulent attempts to gain economic benefits through TK without acknowledging its origin or paying appropriate compensation.

The legal basis for documenting Indonesian TK through IPR is contained in Article 38 of Law No. 28 from 2014 concerning copyright, which contains the State's obligation to document TK. This article states that the State is responsible for documenting, protecting, and maintaining traditional cultural expressions. However, protecting TK through the legal framework of intellectual property rights is not simple. This is mainly because TK is considered public property that has been disseminated and used in traditional communities for a long time [132]. In addition, the owners of TK, namely indigenous peoples, generally

do not have the intention, knowledge, and financial capacity to protect against further claims by outsiders.

Local knowledge has adapted culturally through the centuries and is ingrained in the daily lives of individuals and communities. Naturally, this knowledge is passed down. However, in line with the changing times, which have opened up physical and virtual access and triggered the urbanization of the younger generation and the introduction of modern culture into society, TK will slowly disappear if it is not protected. Incorporating TK into the formal education curriculum and simultaneously strengthening the learning process within the community, especially in the areas known as TK enclaves, are important and strategic efforts that must be carried out by the government. Included in this effort is the development and updating of TK through research. There is an urgent need to increase the intergenerational transmission of local wisdom in line with mainstream education.

## 7. Conclusions

As an archipelagic and multi-ethnic country, Indonesia has a diversity of local wisdom that has a strategic role in watershed management in accordance with the socio-cultural characteristics of local communities. These characteristics are holistically related to the adaptation process in ecosystem management from a sociocultural, economic, and environmental perspective, which has implications for community welfare. Community-based local initiative models for managing watershed ecosystems are still being maintained in various parts of Indonesia. The sustainability of its management simultaneously indicates the continuity of its benefits and value in terms of socio-cultural, economic, and environmental aspects in watershed management and brings a broad impact to people's livelihoods and environmental conservation, especially in terms of the aspects of production, biodiversity conservation, and people's welfare. The diversity of TK models in land resource management is a manifestation of Indonesia's motto "unity in diversity", that is, even though the management models are different, they have one goal of preserving the watershed ecosystem as the main support for people's lives. The government, private sectors, and NGOs cannot work individually to create good watershed conditions, both from a biophysical and socio-economic standpoint. The ideal conditions can only be achieved through the active participation of farmers in adopting and integrating scientific technology into the traditional knowledge they already possess. When modern technology and traditional knowledge (TK) are combined, they can be a powerful tool for assisting in the planning, implementation, monitoring, and evaluation stages of managing natural resources that are vulnerable to degradation, such as those in the management of upstream watersheds. This inclusion is demonstrated in the design and application of LR&SWC approaches, which take into account various aspects, including biophysical, social, economic, and cultural ones. LR&SWC technologies must, however, adhere to the principles of suitability, applicability, feasibility, and acceptability. Additionally, integrating TK into science-based sociotechnical measures uses a variety of communication applications to study, practice, disseminate, and simultaneously maintain the sustainability of local knowledge in the community for watershed management. This is in addition to community groups or network building. It is crucial for watershed management, from planning to monitoring and evaluation, to integrate TK and local wisdom. This will assure the sustainability of long-term watershed management. Documenting TK is necessary to preserve and safeguard indigenous peoples' and local communities' social, cultural, and economic interests. The acknowledgment of IPR goes hand in hand with efforts to preserve TK, which is passed down directly from one generation to the next through formal educational channels, particularly in customary areas and the territories around them.

The slight weakness of this paper is that it only descriptively reviewed traditional knowledge. Quantitative studies supported by spatial analysis will further strengthen the arguments surrounding the existence of traditional knowledge, its positive impacts, and the importance of integrating it into watershed management through a socio-technical approach.

**Author Contributions:** Conceptualization, H.Y.S.H.N., N.W., A.B.S., O.S. and G.N.N.; methodology, H.Y.S.H.N. and N.W.; literature review and data analysis, M.K.S., M.K.A., A.B.S., O.S., G.N.N., D.A., W.I., F.A., L.H. and N.N.N.; writing—original draft preparation, H.Y.S.H.N., M.K.S., M.K.A., N.W., A.B.S., O.S., G.N.N., D.A, W.I., F.A., L.H. and N.N.N.; writing—review and editing, H.Y.S.H.N., M.K.S., M.K.A., N.W., A.B.S., O.S., G.N.N., D.A., W.I., F.A., L.H. and N.N.N. All authors shared roles according to their respective disciplines and experiences as major contributors and equally discussed the conceptual ideas and the outline, provided critical feedback on each section, and helped shape and write the manuscript. All authors have read and agreed to the published version of the manuscript.

**Funding:** This research received no external funding.

**Institutional Review Board Statement:** Not applicable.

**Informed Consent Statement:** Not applicable.

**Data Availability Statement:** Not applicable.

**Acknowledgments:** The authors thank the anonymous reviewers for their useful comments.

**Conflicts of Interest:** The authors declare no conflict of interest.

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
