# Peer review of "Incorporating Traditional Knowledge into Science-Based Sociotechnical Measures in Upper Watershed Management: Theoretical Framework, Existing Practices and the Way Forward"

_sustainability, doi:10.3390/su15043502_

Round 1

Reviewer 1 Report

Indigenous/traditional knowledge is defined as the systematic body of knowledge acquired by local people through accumulation of informal experiences and intensive understanding of their environment in a given society. It is generally accepted that indigenous knowledge (e.g., conserve soil and water resources) can be incorporated into the decision-making process and development activities to enhance watershed management sustainability. As such, this study discusses theoretical framework and implementation practices in integrating traditional knowledge into a science-based sociotechnical system in an effort to manage upstream watersheds in a sustainable manner.

I think this study is a progress by deepening and expanding studies focused on incorporating traditional knowledge into scientific-based sociotechnical measures for watershed management, some revision suggestions are listed below: 1) In Abstract the background texts should be shortened and add your findings. 2) It's better to add a map to illustrate locations in four regions in Indonesia with different characteristics: Nusa Tenggara, Java, Sulawesi, and Kalimantan.

Author Response

Please see the attachhment

Reviewer 2 Report

The paper is an extended descriptive analysis that would benefit in its Introduction of a paragraph that describes the backbone of the paper, the motivation on going on this path and the process behind the Results/Conclusion sections.

In the following text I’ll address the questions needed for the full review. 1. What is the main question addressed by the research?

The main research question is that if there is a correlation between deforestation and low quality of life for nearby population and what is the evolutionary implication.

2. Do you consider the topic original or relevant in the field? Does it address a specific gap in the field?

The paper is relevant to the field, but it needs to have a roadmap or backbone clarification of the paper because the 35 pages are too detailed to keep in mind the main idea and work on it. The paper doesn’t follow a specific gap, it just goes on to create a fundamental overview of the field.

3. What does it add to the subject area compared with other published material?

The material is in line with other papers on the same subject, but its competitive advantage is that it shows a descriptive path of the research and offers a general approach.

4. What specific improvements should the authors consider regarding the methodology? What further controls should be considered?

At this point the paper is just an extend descriptive analysis of the field, but it would benefit from any type of econometric testing on the correlation between deforestation and macroeconomic indicators that highlight the quality of life and the testing should be done first with a lag of one year and then with a lag of two years and evaluate the Errors dispersion and R-squared to understand if the model highlights the reality or it just has academic soundness.

5. Are the conclusions consistent with the evidence and arguments presented and do they address the main question posed?

The Conclusion section hits all marks regarding quality, but would highly benefit from a motivation paragraph regarding the chosen path and also underline the limits of their research especially because of the lack of quantitative testing.

6. Are the references appropriate?

Yes, nothing to comment on this section.

7. Please include any additional comments on the tables and figures.

Nothing to comment on this question.
